# Brain Tumor Growth Inversion via Differentiable Neural Surrogates

**Jonas Weidner**[1,2]                                        *J.WEIDNER@TUM.DE

**Lucas Zimmer**[1]                                          LUCAS.ZIMMER@TUM.DE

**Ivan Ezhov**[1]                                            IVAN.EZHOV@TUM.DE

**Michal Balcerak**[3]                                       MICHAL.BALCERAK@UZH.CH

**Björn Menze**[3]                                           BJOERN.MENZE@UZH.CH

**Daniel Rückert**[1,2,4]                                    DANIEL.RUECKERT@TUM.DE

**Benedikt Wiestler**[1,2]                                   B.WIESTLER@TUM.DE

[1] *Technical University of Munich*

[2] *Munich Center for Machine Learning*

[3] *University of Zurich*

[4] *Imperial College London*

* *Corresponding author*

**Editors:** Accepted for publication at MIDL 2026

## Abstract

Personalizing biophysical brain tumor models to individual patients is computationally expensive due to the need for numerous iterative evaluations of slow numerical solvers to identify optimal patient-specific parameters. We address this by introducing a differentiable neural surrogate that replaces the traditional forward model. Unlike the original solver, this surrogate is fully differentiable, allowing us to solve the inverse problem using highly efficient gradient-based optimization. This approach ensures that the solution learns the biophysical constraints of tumor growth while accelerating the process by orders of magnitude. In a 3D brain tumor growth setting, our framework achieves accuracy competitive with classical optimization while reducing runtime from days to seconds. Crucially, we demonstrate that our method, though trained on synthetic data, generalizes effectively to real patient scans. These findings establish differentiable surrogates as a powerful tool for accelerating scientific machine learning in medical imaging and beyond.[1]

**Keywords:** Neural Surrogates, Gradient-Based Optimization, Brain Tumor Models

## 1. Introduction

Computational modeling provides a central foundation for studying complex biological processes and for enabling patient-specific personalization across many medical applications (Katsoulakis et al., 2024; Kuang et al., 2024; Atad et al., 2025). These models offer mechanistic insight into disease progression, yet adapting them to an individual patient remains computationally demanding. Each personalization step requires numerous evaluations of a slow numerical solver, and a single forward simulation can take several hours. As a result,

---

1. Our code is available at: github.com/jonasw247/brain-tumor-growth-inversion-via-differentiable-neural-surrogates.

estimating patient-specific parameters is a challenging inverse problem (De Domenico et al., 2025). Common optimization strategies, including Bayesian and evolutionary approaches (Weidner et al., 2024a; Lipkova et al., 2019), require repeated simulations and are therefore difficult to use in clinical workflows.

In brain tumor modeling, several approaches have addressed the inverse problem by combining biophysical simulators with Bayesian or evolutionary optimization (Lipkova et al., 2019; Weidner et al., 2024a) or by optimizing discrete losses that encode imaging and physics constraints (Balcerak et al., 2023). These methods achieve clinically meaningful calibrations but still rely on repeated evaluations of a non-differentiable solver and derivative-free search in a parameter space, which limits their runtime and scalability.

Neural surrogates have emerged to mitigate this limitation by approximating simulator input-output mappings with substantial computational speedups (Koehler et al., 2024; Ohana et al., 2024; Salvador et al., 2024; Li et al., 2024; Ezhov et al., 2021; Alkin et al., 2025). However, they typically focus on forward predictions rather than solving inverse problems. Most of these approaches are used as fast forward emulators, but they are rarely embedded into a fully differentiable inverse calibration loop. While Physics-Informed Neural Networks (PINNs) (Cuomo et al., 2022; Garay et al., 2024) address the inverse setting, the competing objectives of data-fitting and physical consistency often lead to convergence failure, particularly when modeling sharp interfaces or discontinuities common in biological tissues (Zhang et al., 2025). The most recent optimization approaches that embed physics as soft constraints (Balcerak et al., 2023; Karnakov et al., 2022) face similar limitations and still do not sufficiently amortize the computational cost. They effectively require training a fresh optimizer for every new patient geometry.

We introduce a fresh approach for solving the inverse problem in personalized medical simulations by turning a non-differentiable biophysical tumor growth model into a fully differentiable system through a neural surrogate, which bridges numerical modeling and gradient-based optimization.[2] Our core contributions, therefore, are:

**(i)** We introduce a differentiable neural surrogate for inverse calibration of partial differential equations (PDE) - based tumor growth models, achieving a speedup from days to seconds.
**(ii)** We provide a detailed analysis of failure modes, parameter space limitations, and out-of-distribution behavior, including evaluation on real patient data.
**(iii)** Our findings suggest that differentiable neural surrogates provide a practical foundation for scalable and precise personalization of biophysical models, with the potential to extend to other clinical applications beyond brain tumors.

## 2. Methods

We aim to personalize biophysical tumor models by fitting simulated tumor concentrations to individual patients' magnetic resonance images (MRI). Accurate personalization is essential for radiotherapy planning, particularly to infer tumor infiltration in regions that are not visible in the images. By estimating patient-specific tumor growth coefficients, we seek

---

2. This manuscript extends a previously published short paper (Weidner et al., 2024b) by adding comparative optimization strategies, a detailed failure analysis, a high precision forward run in the end, and an explicit hybrid initialization.

to optimize treatment targets beyond the radiologically defined tumor boundaries. The complete problem formulation is described in Figure 1.

Typically, the final stage of personalization introduces an imaging function that transforms the continuous tumor-cell concentration into a binary segmentation for comparison with MRI. Earlier studies tested several hand-crafted imaging functions and calibrated some of their parameters (Weidner et al., 2024b; Balcerak et al., 2023; Lipkova et al., 2019; Weidner et al., 2024a). To remove this additional source of uncertainty and focus squarely on the inverse problem of calibrating the biophysical model, we perform parameter estimation directly against the simulated "ground-truth" concentration.

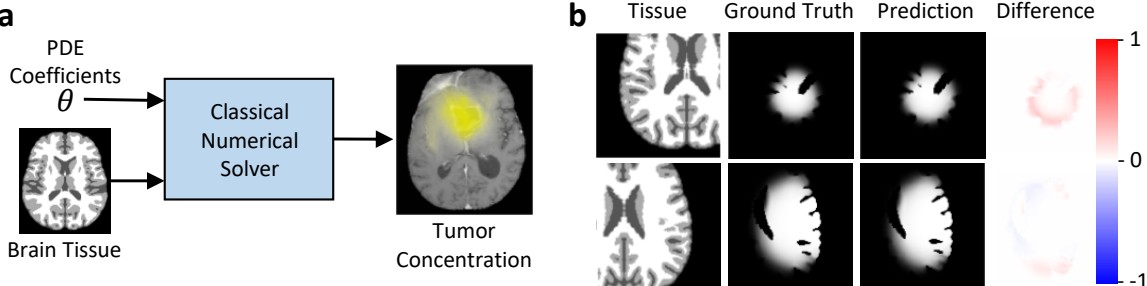

Figure 1: **(a)** Our goal is to estimate the invisible 3-D brain tumor concentration based on the visible tumor segmentation on the MRI and biophysical constraints. To achieve this, we simulate tumor growth by inputting the PDE coefficients and brain tissue into the numerical solver, which predicts the tumor concentration. Optimizing those coefficients using classical methods is slow, thus limiting the clinical adaptation of complex brain tumor models. **(b)** We introduce a precise and differentiable neural surrogate enabling fast personalization. The predictions of this surrogate are displayed next to the ground truth of the numerical solver.

## 2.1. Biophysical Brain Tumor Modeling

In this study, we focus on the most common primary malignant brain tumor, IDHwt glioblastoma, which is characterized by its diffuse infiltration and has been extensively studied in prior work, e.g. (Ezhov et al., 2023; Balcerak et al., 2023; Lipkova et al., 2019). As a biophysical model, we apply the widely used Fisher-Kolmogorov equation for brain tumor growth:

$$\frac{\partial c(\mathbf{x}, t)}{\partial t} = \nabla \cdot \big( D(\mathbf{x}) \, \nabla c(\mathbf{x}, t) \big) \; + \; \rho \, c(\mathbf{x}, t) \, \big( 1 - c(\mathbf{x}, t) \big), \tag{1}$$

where $c(\mathbf{x}, t)$ is the tumor-cell density, $D(\mathbf{x})$ the tissue-dependent diffusion coefficient, and $\rho$ is the proliferation rate. The brain volume is segmented into white matter (WM), gray matter (GM) and cerebrospinal fluid (CSF) resulting in tissue-specific diffusion: $D_{\mathrm{GM}} = 0.1 \, D_{\mathrm{WM}}, D_{\mathrm{CSF}} = 0$ as tumor cells infiltrate GM more slowly and are assumed not to invade CSF. The forward solver inputs the proliferation rate $\rho$, diffusion coefficient $D_{\mathrm{WM}}$, seed location $(x, y, z)$, and total growth time $T$, i.e. the parameter set $\theta_{\mathrm{orig}} = \{x, y, z, \rho, D_{\mathrm{WM}}, T\}$.

Typically, only a single time point is available for each patient, and the start time of the tumor growth process is unknown. Thus, $\theta_{\text{orig}}$ is not identifiable. For example, a given tumor could have grown fast over a short period of time ($\rho \uparrow$, $T \downarrow$) or slow, but over a longer duration ($\rho \downarrow$, $T \uparrow$). Following (Ezhov et al., 2023), we unify $T$ into two scale-free parameters $\mu_D = \sqrt{D_{\text{WM}} T}, \mu_\rho = \sqrt{\rho T}$, and optimise the reduced set $\theta = \{x, y, z, \mu_D, \mu_\rho\}$. Given $\theta$, the numerical solver returns the 3-D tumor cell-density field $c(\mathbf{x}; \theta)$.

## 2.2. Numerical Solver

We use the open source TumorGrowthToolkit (Balcerak et al., 2023) to simulate the Fisher-Kolmogorov equation (Equation 1) and generate synthetic data, which was already applied in several previous studies (Haouari et al., 2025; Weidner et al., 2025; Balcerak et al., 2024). The solver operates on a Cartesian grid using an explicit finite-difference scheme with tissue-dependent diffusion. The diffusion term is discretized in flux form with face-centered diffusion coefficients derived from white- and gray-matter maps, while time integration is performed using a forward Euler update. The time step is chosen conservatively to satisfy a Courant–Friedrichs–Lewy type stability condition for the diffusion term and to prevent overshooting of the reaction term.

## 2.3. Neural Surrogate Model

First, we train a neural surrogate that approximates the forward solver of the reaction–diffusion PDE. During training, the network receives the PDE coefficients as input together with the relevant side constraints, such as the initial tumor seed, tissue mask, and boundary conditions. The output is the final tumor-cell concentration obtained by the numerical solver. The surrogate learns the explicit mapping $f_\phi : (\theta, \mathcal{B}) \longmapsto c(\mathbf{x})$, where $\theta$ denotes the set of biophysical coefficients, $\phi$ denotes the surrogate weights and $\mathcal{B}$ the boundary data. Because the forward problem is well-posed, this supervised learning task is stable and admits a unique solution, orders of magnitude faster than the classical solver. Empirically, U-Net architectures have proved effective for this task (Haouari et al., 2025; Ronneberger et al., 2015). Recent studies further show that substituting the standard convolutional blocks in a U-Net with ConvNeXt blocks yields superior performance on reaction–diffusion problems (Liu et al., 2022; Ohana et al., 2024). We therefore adopt this ConvNeXt-U-Net variant. A comparison to other forward models can be found in the Appendix B. The PDE coefficients are first processed by a fully connected layer, and the resulting activations are added channel-wise to the U-Net bottleneck. Injecting the coefficients at this most abstract level allows the latent representation to modulate spatial features in a physics-informed manner.

We use a 3D ConvNeXt–U-Net with 4 stages and one block per stage. Downsampling and upsampling are done with a kernel size of 2 and a stride of 2. We train the neural surrogate using the Adam optimizer with a learning rate of $10^{-4}$ and a batch size of 3 for 35 h. The PDE coefficients ($\mu_\rho$, $\mu_D$, and seed coordinates) are processed through a fully connected layer and injected into the bottleneck. The output layer uses a sigmoid activation to constrain predictions to $[0, 1]$, representing normalized tumor cell concentration. A brain mask is applied post-prediction to zero out concentrations outside the brain. We optimized a

mean squared error (MSE) loss between the predicted and ground-truth tumor concentration fields.

## 2.4. Inverse Model

For inverse optimization, we evaluate the following approaches.

**CMA-ES classical solver:** As a classical baseline, we employ the CMA-ES (`cma` Python package) to optimize the tumor-concentration model (Hansen and Ostermeier, 2001). This represents the standard evolutionary approach where a population of candidate solutions is iteratively evolved using the slow, traditional numerical solver.

**CMA-ES neural surrogate:** Analogously, we also run CMA-ES directly with the neural surrogate. In this setting, the expensive numerical solver is replaced by the surrogate, yet optimization remains entirely derivative-free. The gradients of the network are intentionally ignored. This ablation allows us to assess how much of the performance gain stems from the surrogate's speed alone versus the use of gradient-based inversion.

**Direct inverse prediction model:** An obvious solution is the prediction of the coefficients directly by a network (Figure 2a). For this approach, we used a ConvNeXt (Liu et al., 2022) architecture similar to the encoder of the forward neural surrogate model. The network inputs the tumor concentration and the brain tissue geometry to predict the PDE coefficients.

**Gradient-based (GB) neural surrogate:** We take a neural surrogate trained until convergence and freeze its weights to exploit its end-to-end differentiability for inverse optimization. Treating $f_\phi$ as a differentiable function of $\theta$, we compute the exact gradient of the $\ell_2$ loss between the predicted and observed tumor maps. This gradient is then supplied to a memory-efficient quasi-Newton optimiser (L-BFGS) that iteratively updates $\theta$ to minimise the loss (Figure 2b and Algorithm 1).

**Inverse Prediction and GB Optimization:** We integrate the two complementary inversion strategies, direct inverse prediction and the gradient-based neural surrogate, into a single hybrid pipeline. Concretely, we first execute the direct inverse model to obtain a coarse but physically plausible estimate $\theta_{\mathrm{DI}}$ of the PDE coefficients. This estimate is then used to initialise the gradient-based optimiser that operates on the neural surrogate. Starting from $\theta_{\mathrm{DI}}$ provides the quasi-Newton solver with a point already close to the true optimum, reducing the risk of entrapment in poor local minimum. After the GB optimisation terminates, we evaluate both candidate solutions, the original direct-inverse estimate and the refined GB surrogate estimate. The parameter set that achieves the lower loss is retained as the *final* prediction. This two-stage procedure combines the speed and robustness of direct inversion with the accuracy of gradient-based fine-tuning.

For the real patient experiments, we add additional baselines that operate directly on the acquired patient images and tumor segmentations.

**GliODIL**: Here, the full tumor growth dynamic over time is optimized. The discrete PDE loss is optimized across a multilevel regular grid, and a separate data term penalizes deviations between the simulated and observed tumor states at the initial and final time points. The final prediction is generated with a forward run of the numerical solver with the optimized coefficients (Balcerak et al., 2023).

---

**Algorithm 1:** Gradient-based inversion with a differentiable neural surrogate (L-BFGS)

---

**Input:** Frozen surrogate $f_\phi$, boundary/tissue $B$, target tumor $c^\star$, initial coefficients $\theta_0$, stopping condition $\epsilon$

**Output:** Estimated coefficients $\hat{\theta}$

$\theta \leftarrow \theta_0$            `// initialize coefficients`

$L_{\mathrm{prev}} \leftarrow 0$            `// initialize previous loss`

$L \leftarrow +\infty$            `// initialize current loss`

**while** $|L_{\mathrm{prev}} - L| > \epsilon$ **do**

    $L_{\mathrm{prev}} \leftarrow L$       `// store loss for stopping check`

    $\hat{c} \leftarrow f_\phi(B, \theta)$       `// forward surrogate evaluation`

    $L \leftarrow \mathrm{MSE}(\hat{c}, c^\star)$       `// compute loss`

    $\nabla_\theta L \leftarrow \frac{\partial L}{\partial \theta}$       `// backpropagate gradients`

    $\theta \leftarrow \mathrm{L\text{-}BFGS}_{\mathrm{step}}(\theta, \nabla_\theta L)$       `// L-BFGS update using stored curvature`

**end**

$\hat{\theta} \leftarrow \theta$            `// final estimated coefficients`

---

**LMI**: The PDE coefficients are estimated with a direct inversely trained network similar to the direct inverse prediction model we used. In contrast to our direct inverse prediction approach, LMI registers the patient's anatomy into an atlas, runs inference there, and then registers back into patient space (Ezhov et al., 2023).

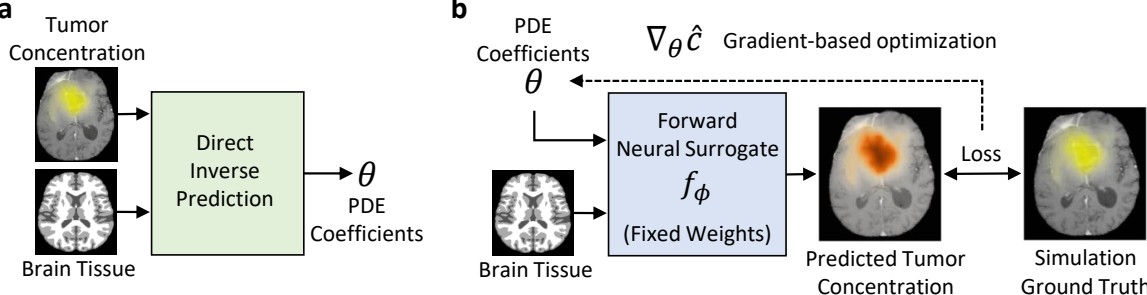

Figure 2: Overview of our neural surrogate optimization. The goal is to estimate the optimal PDE coefficients that explain the brain tumor. **(a)** The direct prediction model estimates the PDE coefficients that explain the given tumor concentration, thus it solves the inverse prediction within one inference run. **(b)** A differentiable neural surrogate model is trained on the forward tumor simulation problem. It inputs the PDE coefficients and the brain tissue and outputs a tumor concentration. Finally, the patient-specific coefficients are estimated using gradient-based optimization.

## 2.5. Metrics

We evaluate surrogate- and solver-generated (ground truth) tumor fields with five complementary metrics. Mean-squared error (MSE) captures the total energy of numerical

deviations and harshly penalizes large local mistakes, while mean absolute error (MAE) reports the voxel-wise bias in the same physical units as the data. Because raw errors scale with lesion size, we also report volume-normalized MSE and MAE, dividing each by the ground-truth tumor volume so results remain comparable across samples with different tumor extents. Normalized cross-correlation (NCC) measures how well the spatial pattern of low- and high-concentration regions aligns, independent of any global scale or offset. This metric is particularly interesting for radiotherapy planning, which acts on the relative radiation distribution. For the real patient examples, we also evaluate the Dice score against the true tumor segmentation. As we do not want to focus on a specific imaging function for mapping from continuous tumor concentration to a binary mask, we use a simple threshold of 0.33, the same one used within the imaging function of the simulation, which serves as our target.

## 2.6. Data

**Synthetic Data:** We used a dataset, designed to match tumor sizes seen in the BRATS dataset (Menze et al., 2014). We trained on 28,000 samples, validated on 1,000, and tested on 500. Due to cost constraints, we evaluated the full-resolution CMA-ES only on a subset of 25. The details are described in Appendix D.

**Patient Data:** We evaluate our model without fine-tuning on 75 randomly selected patients from the BraTS dataset (release 2021, (Baid et al., 2021)). Tissue segmentation into white matter, gray matter, and cerebrospinal fluid was obtained by deformable atlas registration (ANTs SyN) (Avants et al., 2009) to the patient's imaging space, instead of conventional segmentation tools that perform poorly in tumor regions. This registration-based approach aims to estimate healthy brain anatomy under tumor-induced mass effect and is inspired by the ideas presented in (Scheufele et al., 2020). As there is no ground truth available for real patients, we use the best estimate, the CMA-ES solver, with a classical numerical solver as ground truth and test our models against it. In addition, we calculate the Dice overlap of thresholded tumor cell concentration maps with the observed tumor segmentation.

**Preprocessing:** Each 3-D scan is cropped to a $128 \times 128 \times 128$ voxel cube centred on the tumor's center of mass. Because all samples are already registered to a common anatomical orientation, we omit any rotation-based augmentation. We used an NVIDIA Quadro RTX 8000 and an AMD EPYC 7313 16-Core Processor for all experiments.

## 3. Results

We first validate the neural surrogate's forward-solving accuracy. We then assess the ability to recover the underlying PDE coefficients in the inverse problem using multiple approaches and, finally, examine how the recovered parameters affect the predicted tumor concentration. Additionally, we evaluate how our models perform on real patients.

## 3.1. Forward neural surrogate

Across the test set, the surrogate approximates the numerical solver over the full span of physiologically plausible parameters, achieving a mean-squared error of MSE $= 1.8 \times 10^{-3}$. This level of accuracy is consistent with the findings of previous works (Haouari et al.,

2025). Qualitative side-by-side comparisons between surrogate predictions and ground-truth simulations are provided in Figure 1b. These results show that the forward model is sufficiently precise for the subsequent inversion experiments, with errors comfortably within the clinically acceptable margin, especially when one accounts for the inherent biological uncertainties. A stability analysis is conducted in the Appendix A.

### 3.2. Coefficient prediction

Table 1: PDE coefficient prediction. We compare the prediction of the tumor growth parameter $\mu_\rho$, the diffusion coefficient $\mu_D$, and the tumor origin to the ground truth values.

| Model | Growth MSE ($\downarrow$) ($\times 10^{-3}$ %) | Growth MAE ($\downarrow$) ($\times 10^{-3}$ %) | Diffusion MSE ($\downarrow$) ($\times 10^{-3}$ %) | Diffusion MAE ($\downarrow$) ($\times 10^{-3}$%) | Origin MSE ($\downarrow$) ($\times 10^{-3}$) | Origin MAE ($\downarrow$) ($\times 10^{-3}$) |
|---|---|---|---|---|---|---|
| CMA-ES Classical Solver | 68.3±30.2 | 177.7±38.3 | 122.8±36.4 | 272.2±44.1 | 17.7±10.0 | 111.8±41.7 |
| CMA-ES Neural Surrogate | 16.5±3.5 | 62.8±5.0 | **53.6±12.1** | 112.3±9.1 | 9.6±10.5 | 72.2±38.4 |
| Direct Inverse Prediction | 32.8±4.2 | 124.1±5.9 | 295.4±282.7 | 93.9±23.9 | **6.8±4.3** | 72.6±22.5 |
| GB Neural Surrogate | **11.7±3.2** | **52.1±4.2** | 279.4±273.0 | **70.8±23.4** | 8.1±8.7 | **66.9±34.9** |
| CMA-ES Classical Solver (median) | 12.2 | 110.2 | 72.4 | 269.2 | 14.3 | 119.6 |
| CMA-ES Neural Surrogate (median) | 0.8 | 27.5 | 1.6 | 40.5 | 2.4 | 49.4 |
| Direct Inverse Prediction (median) | 8.3 | 90.8 | 2.4 | 48.8 | 4.2 | 65.1 |
| GB Neural Surrogate (median) | **0.7** | **27.0** | **1.2** | **34.9** | **2.1** | **45.5** |

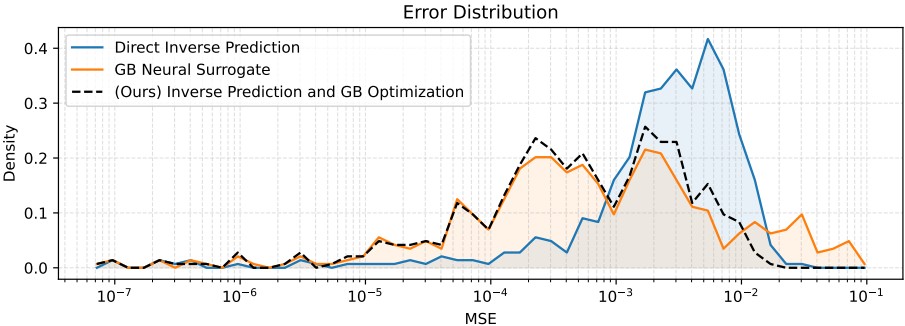

Figure 3: Error distribution for the different methods. Optimization approaches and direct inverse prediction show good performance in different scales, which makes the combination of both valuable. We find that the direct prediction generally provides solid results ($1 \times 10^{-3}$ - $1 \times 10^{-2}$), but lacks really good results. The optimization approaches can reduce the error by a further order of magnitude for most cases, which is crucial for clinical applications. In some cases, the forward optimization fails drastically ($1 \times 10^{-2}$ - $1 \times 10^{-1}$). Using both combined (dashed) drastically improves the result (Table 2).

We evaluate the different approaches based on their ability to predict the ground-truth PDE coefficients. The results are shown in Table 1. We report the mean and its standard

deviation alongside the median to provide a balanced performance profile: The mean value reflects the overall risk profile, including rare but costly errors, while the median provides a robust measure. For the neural surrogate method, the mean errors are inflated by a handful of runs that converge to incorrect solutions, whereas the typical error, as captured by the median, remains significantly lower. The direct inverse prediction and the GB Neural Surrogate approaches exhibit the same pattern, with their means inflated by occasional collapses, even though most trials yield low error.

Our quantitative experiments reveal clear differences across optimisation strategies. The classical numerical solver, optimized with CMA-ES, exhibits consistently large errors in the recovered PDE coefficients. The purely data-driven direct inverse network fares slightly better. It struggles to disentangle the ill-posed inverse mapping and therefore fails to reproduce the ground-truth coefficients with acceptable accuracy. In contrast, both flavours of our surrogate-based optimisation, the gradient-based variant and the derivative-free CMA-ES variant, converge to markedly lower errors. But the gradient-based approach shows slight advantages. The mean error, however, remains noticeably larger owing to a small number of outlier cases in which optimisation becomes trapped in secondary minima. An example is studied in detail in the Appendix (Figure 5).

In summary, classical CMA-ES optimisation and direct prediction alone are inadequate, whereas the neural surrogate combined with either gradient-based or CMA-ES optimisation delivers robust and accurate parameter recovery, with a modest edge for the gradient-based approach.

### 3.3. Final Tumor Prediction

The goal of our approach is to determine the optimal set of coefficients that describes the patient's tumor, thereby obtaining a matching tumor concentration. The results are shown in Table 2. The classical pipeline that couples the numerical solver with CMA-ES attains a low simulation error, even though we found that the recovered coefficients deviate substantially from the ground truth. This apparent contradiction arises from the ill-posed nature of the inverse problem: multiple parameter combinations can reproduce the observed tumor distribution similarly well, enabling the optimiser to explain the data without converging to the true coefficients.

The results of the surrogate-based methods and direct inverse prediction show worse performance compared to the classical solver.

By investigating the error distribution (Figure 3), we find that the GB Neural Surrogate shows a small error on a lot of samples but fails for some cases. A few failed optimization runs dominate the mean values. In contrast, the direct inverse prediction has no real failure cases but typically has a larger error. By applying our hybrid approach, which combines both optimization approaches, we can strongly reduce those failure cases. This results in significantly better performance, even outperforming classical optimization on the clinically relevant NCC metric, which highlights the relative differences.

### 3.4. Real patients

Although we trained on synthetic cases in an atlas, our approach is also applicable to real patient data with unseen geometry. As shown in Figure 4, it produces anatomically plausible

Table 2: Main findings. Performance comparison for the final forward run with a classical solver following strict physical constraints. We compare mean squared error (MSE), mean absolute error (MAE), also normalized by the ground truth tumor volume (MSE / MAE Normalized), the normalized cross correlation (NCC), and the runtime. We report the mean and the standard error. Bold indicates best, underlined indicates second best. Significant differences (p < 0.01, paired t-test) to our method are marked with (*).

| Model | MSE (↓) (×10⁻³) | MAE (↓) (×10⁻³) | MSE norm. (↓) (×10⁻⁸) | MAE norm. (↓) (×10⁻⁸) | NCC (↑) (×10⁻¹) | Runtime (↓) (min) |
|---|---|---|---|---|---|---|
| CMA-ES classical solver | **0.34 ± 0.08** | **1.55 ± 0.24** | **0.77 ± 0.22** | **4.00 ± 0.70** | **8.95 ± 0.37** | 2300 |
| CMA-ES Neural Surrogate | 5.42 ± 0.59* | 8.10 ± 0.65* | 4.59 ± 0.43* | 8.07 ± 0.47* | 7.81 ± 0.13* | 2.5 |
| Direct Inverse Prediction | 4.81 ± 0.20* | 8.87 ± 0.32* | 5.02 ± 0.23* | 10.08 ± 0.40* | 8.47 ± 0.04* | **0.1** |
| GB Neural Surrogate | 5.91 ± 0.65* | 8.53 ± 0.73* | 4.88 ± 0.48* | 7.99 ± 0.53* | 8.19 ± 0.11* | 0.7 |
| (Ours) Inverse Prediction and GB Optimization | 1.35 ± 0.19 | 3.29 ± 0.29 | 1.36 ± 0.15 | 4.05 ± 0.25 | 9.15 ± 0.10 | 0.8 |

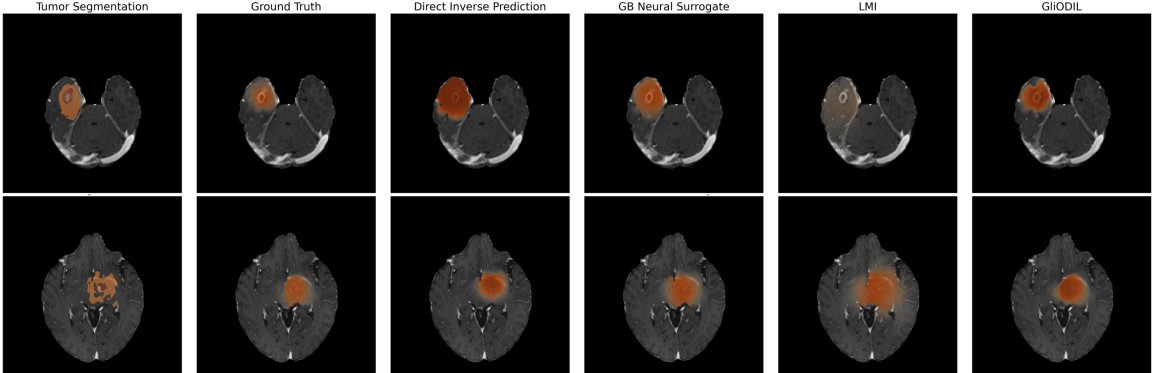

Figure 4: Illustrative application to two clinical BraTS patients. The inferred tumor-cell concentration (axial slice) reveals infiltrative spread well beyond the radiological segmentation, highlighting tumor tissue that would be missed by standard radiotherapy planning. We compare all methods against the ground truth simulation, which represents our best estimate of the underlying tumor concentration. This reference is obtained via a computationally expensive inference pipeline that combines CMA-ES with a high-precision numerical solver and an imaging model fitted to the MRI segmentation. We observe clear discrepancies between the ground truth and both direct inversion and LMI, whereas GliODIL and our inverse neural approach closely match the true concentration.

infiltration patterns on real patient scans. We evaluate our model using a set of real patient data (Table 3). The direct inverse prediction results show a lower MSE and higher NCC than the GB neural surrogate model. We attribute this to the robustness of the direct prediction, as discussed in Figure 3. Additionally, we find that the GB neural surrogate occasionally struggles to predict the correct origin. Thus, we added another run, fixing the origin to the

center of mass, which results in a clear improvement. As in the synthetic case, our combined approach significantly outperforms the individual methods, showcasing the complementarity of the presented methods.

When comparing our approach with GliODIL and LMI, we observe that GliODIL attains a marginal, statistically non-significant reduction in MSE, while its NCC is lower and its runtime is longer by more than an order of magnitude. Overall, GliODIL therefore does not provide a clear accuracy benefit over our method, but it comes at a substantially higher computational cost, which limits its practical applicability in a clinical setting. LMI, in turn, consistently underperforms our direct inverse prediction and reaches only similar accuracy to the GB neural surrogate. We hypothesize that this degradation is mainly driven by errors introduced during the required registration of patient geometries.

Table 3: Fit to real patients relative to the CMA-ES reference. As the neural surrogate on an atlas brain, the model struggles to estimate the tumor origin correctly. Thus, we ran another forward run with the tumor center of mass (COM) as origin. We report the mean and the standard error. Bold indicates best, underlined indicates second best. Significant differences (p < 0.01, paired t-test) to our method with COM are marked with (*).

| Model | MSE (↓) ($\times 10^{-3}$) | MAE (↓) ($\times 10^{-3}$) | MSE norm. (↓) ($\times 10^{-8}$) | MAE norm. (↓) ($\times 10^{-8}$) | NCC (↑) ($\times 10^{-1}$) | Dice (↑) ($\times 10^{-2}$) | Runtime (↓) (min) |
|---|---|---|---|---|---|---|---|
| LMI | 5.16 ± 0.98* | 12.98 ± 1.53* | 8.90 ± 1.69* | 24.21 ± 3.15* | 7.41 ± 0.21* | 42 ± 02 | 5 |
| GliODIL | **1.76 ± 0.24** | **4.82 ± 0.46*** | **2.18 ± 0.19** | **6.75 ± 0.40** | 8.53 ± 0.16* | **61 ± 2** | 50 |
| Direct Inverse Prediction | 2.83 ± 0.38* | 7.14 ± 0.79* | 3.60 ± 0.32* | 9.38 ± 0.56* | 8.93 ± 0.12* | 57 ± 1 | 0.1 |
| GB Neural Surrogate | 5.76 ± 0.69* | 11.13 ± 1.12* | 8.01 ± 0.59* | 16.21 ± 0.94* | 4.81 ± 0.41* | 30 ± 3 | 0.7 |
| GB Neural Surrogate COM | 4.83 ± 1.13* | 10.32 ± 1.68* | 4.24 ± 0.65* | 11.29 ± 0.93* | 9.16 ± 0.09* | 55 ± 1 | 0.7 |
| (Ours) Inverse Prediction and GB Optimization | 2.47 ± 0.36 | 6.51 ± 0.75 | 3.14 ± 0.28 | 8.62 ± 0.50 | 8.83 ± 0.14 | 34 ± 2 | 0.8 |
| (Ours) Inverse Prediction and GB Optimization COM | 2.10 ± 0.36 | 6.24 ± 0.75 | 2.21 ± 0.24 | 7.76 ± 0.48 | **9.42 ± 0.05** | 59 ± 1 | 0.8 |

## 4. Discussion

We find that differentiable neural surrogates can drastically accelerate the personalization of glioma simulations. The inferred model parameters have meaningful clinical interpretations. In Fisher-Kolmogorov type growth models fitted to preoperative MRI, diffusion and proliferation coefficients have been proposed as patient-specific biomarkers. It has been shown that these estimated coefficients reflect the balance between tumor invasion and growth, and they as well as the simulated tumor burden from these estimated parameters, correlate with overall survival, while proliferation-related parameters have been linked to underlying molecular pathway activity (Metz et al., 2024). This highlights the potential of our framework not only to generate anatomically plausible tumor cell maps but also to support clinically relevant prognosis.

Additionally, prior work has demonstrated that estimating tumor concentration using biophysical growth models can support and improve radiotherapy treatment planning (Balcerak et al., 2023) by defining personalized clinical target volumes (CTV).

We have demonstrated that learning the well-posed forward path and gradient-based optimization is advantageous compared to directly predicting the ill-posed problem. However, occasional optimization runs converged to a suboptimal local minimum, which impacted the mean error metrics. Our stability analysis confirms that convergence reliability decreases substantially with larger initialization perturbations, motivating our hybrid strategy where direct inverse prediction provides initialization already within the convergent basin. We combined gradient-based optimization with direct inverse prediction. Thus, we achieved a speedup from days to seconds while improving the quantitative metrics such as normalised cross-correlation. The latter focuses on the relative difference to the ground truth, which is crucial for the subsequent application of the resulting tumor cell maps to radiotherapy planning. Our results on real patient data suggest that the surrogate enables anatomically plausible infiltration patterns even though it was only trained on atlas anatomy. The inversion process shows sensitivity to the predicted tumor origin, likely due to anatomical variability not present in the synthetic training distribution. Initializing the optimizer with the tumor center of mass reduces this effect, highlighting the need for surrogate models trained on a broader set of anatomies to improve robustness.

Out-of-distribution samples remain challenging, especially when the PDE coefficients lie near the edges of the training domain, where we observe more frequent failures. Expanding the coefficient ranges during training could reduce these boundary effects. Another way to increase performance would be to subsequently sample over the coefficient space using a traditional optimization approach, with the gradient-based optimization result serving as prior or initialization, as shown in (Weidner et al., 2024a). Another limitation of the proposed approach is its reliance on a large synthetic training dataset that closely matches the real data distribution. As a result, a substantial portion of the computational cost is shifted to the surrogate training phase. While this upfront cost is beneficial for applications that require repeated inference, such as patient-specific brain tumor modeling, it may be less practical for use cases with only a few runs or highly specialized simulation settings.

In future work, we will focus on enhancing optimization robustness and extending our framework to further medical simulation tasks, including more complex biophysical models with additional parameters, such as anisotropy or mass effect (Subramanian et al., 2022; Bortfeld and Buti, 2022), which are currently difficult to optimize due to their computational cost. Especially for high-dimensional problems, we expect gradient-based optimization to provide a decisive advantage.

We believe that differentiable neural surrogates provide a practical foundation for scalable and precise personalization of biophysical models. Their efficiency opens new opportunities for integrating mechanistic modeling into clinical pipelines and for exploring patient-specific dynamics in ways that were previously computationally inaccessible.

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

## Appendix A. Stability Analysis

### A.1. Surrogate Stability

We investigate the stability of the forward neural surrogate in Figure 5. We showcase the impact of changing initial conditions on the performance.

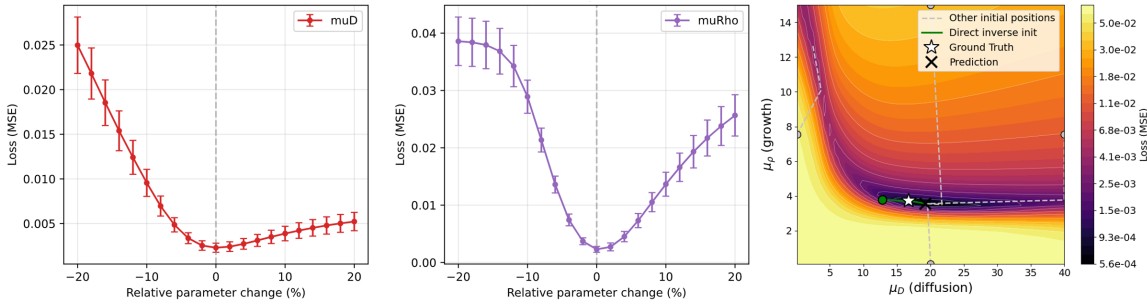

Figure 5: Stability analysis. We assess how sensitive the forward surrogate loss is to perturbations of the inferred parameters. Left and middle, we vary the diffusion coefficient $\mu_D$ and the proliferation coefficient $\mu_\rho$ by $\pm 20\%$ (x axis) while keeping all other quantities fixed, and report the resulting MSE, with error bars indicating variability across patients. Underestimation (negative perturbations) increases the loss more strongly than overestimation, and the objective is overall more sensitive to $\mu_\rho$. Right, loss landscape for one representative sample over $(\mu_D, \mu_\rho)$ shows a similar trend. Increased proliferation and diffusion coefficients lead to increased tumor volumes. This is the reason for the visible loss valley. We overlay optimization trajectories starting from our direct inverse initialization (green) and additional initializations (gray), most runs converge to the same basin, while one converges to a distinct local minimum.

### A.2. Inverse Optimization Stability

We evaluate the sensitivity of the inverse optimization to different parameter initializations. Therefore, we vary the radius of the initialization region around the reference parameters and assess convergence behavior relative to the ground truth. In this analysis, we exclusively study the stability of the diffusion coefficient and proliferation rate, while keeping the tumor origin initialization fixed. For each radius, parameters are sampled uniformly at random within a specified range and evaluated over 190 samples.

Table 4 summarizes the results. For small perturbations of up to 10% in the parameter initialization, approximately 80% of runs converge to the same minimum, which we define as a deviation of less than 20% in the inferred parameters. In contrast, when increasing the initialization range to 30%, only about 50% of runs converge to the same minimum. A similar trend is observed for both the deviation from the ground truth and the deviation from the prediction at zero radius. Overall, initialization variability of up to 10% results in stable

and consistent inverse optimization outcomes, whereas larger perturbations substantially increase the outcome variability. An example of such a failure mode can be found in Figure 5 (right), where one of the optimization runs converges to a local minima.

Overall, these results indicate that a well-chosen initialization improves the stability and reliability of inverse optimization.

Table 4: Comparison of different initializations. We sample in a radius around the coefficient space. The sample radius percentage is measured in comparison to the full parameter space. We measure the deviation of the optimized coefficients compared to the ground truth of the numerical solver and to the optimization result of the radius 0, our initialization based on the direct inverse prediction. Additionally, we show the ratio of runs that converge to the same minimum as our initialization.

| Sample Radius | Ground Truth Difference $\mu_D$ ($\downarrow$) | Ground Truth Difference $\mu_\rho$ ($\downarrow$) | Prediction Difference $\mu_D$ ($\downarrow$) | Prediction Difference $\mu_\rho$ ($\downarrow$) | Same Minimum Ratio |
|---|---|---|---|---|---|
| 0% | $0.72 \pm 0.08$ | $0.17 \pm 0.02$ | - | - | - |
| 1% | $1.16 \pm 0.13$ | $0.24 \pm 0.03$ | $1.28 \pm 0.13$ | $0.24 \pm 0.03$ | 89.5% |
| 10% | $1.53 \pm 0.17$ | $0.35 \pm 0.04$ | $1.58 \pm 0.16$ | $0.31 \pm 0.04$ | 79.5% |
| 30% | $3.25 \pm 0.26$ | $1.02 \pm 0.11$ | $3.37 \pm 0.24$ | $1.03 \pm 0.11$ | 52.1% |

## Appendix B. Comparison Forward Surrogate Architecture

We include a targeted study to assess the influence of the forward surrogate architecture. We compare a vanilla U-Net (Ronneberger et al., 2015), U-Net++ (Zhou et al., 2018), V-Net (Milletari et al., 2016), and the ConvNeXt–U-Net (Liu et al., 2022) used in our main experiments. All architectures are trained under identical conditions to approximate the same numerical tumor growth solver. For this study, we limit the training time of all models to 24 hours and 30GB of Memory. The batch size was adapted to fit into memory. Additionally, we reduced the training set size to 3000 samples.

We focus on forward accuracy, as sufficiently low approximation error is the primary requirement for reliable gradient-based inversion. As shown in Table 5, all architectures achieve comparable performance except for the U-Net++, which is slightly inferior across most metrics. The ConvNeXt–U-Net exhibits barely improved normalized metrics, while the VNet performs slightly better on absolute MSE and MAE. VNet and ConvNeXt–U-Net perform best on NCC. Overall, these results suggest that forward accuracy is comparable across architectures and sufficiently high for reliable inverse optimization, indicating that the specific backbone choice plays a minor role in the proposed framework. For comparisons with non-U-Net architectures, we refer to previous work (Haouari et al., 2025).

## Appendix C. Error Analysis

In the following, we analyze the failure cases of the GB Neural Surrogate approach. We visualize the mean squared error (MSE) in Figure 6. We observe that large errors occur in regions with high normalized growth rates combined with high normalized diffusion coefficients. This initially suggested that the model struggles with larger tumors. However,

Table 5: Forward model study. We evaluate MSE, MAE, and the relative version, which divides by the ground truth volume. The best row is indicated in bold and the second best is underlined.

| Model | MSE ($\downarrow$) ($\times 10^{-3}$) | MAE ($\downarrow$) ($\times 10^{-2}$) | MSE norm. ($\downarrow$) ($\times 10^{-8}$) | MAE norm. ($\downarrow$) ($\times 10^{-7}$) | NCC ($\uparrow$) ($\times 10^{0}$) |
|---|---|---|---|---|---|
| U-Net | $2.20 \pm 0.12$ | $1.04 \pm 0.03$ | $\underline{3.63 \pm 0.48}$ | $4.01 \pm 0.80$ | $0.9413 \pm 0.0059$ |
| U-Net++ | $3.17 \pm 0.17$ | $1.26 \pm 0.04$ | $5.88 \pm 1.06$ | $3.76 \pm 0.60$ | $0.9112 \pm 0.0081$ |
| V-Net | $\mathbf{1.72 \pm 0.10}$ | $\mathbf{0.85 \pm 0.03}$ | $8.28 \pm 5.63$ | $\underline{3.22 \pm 1.08}$ | $\mathbf{0.9493 \pm 0.0059}$ |
| ConvNeXt-U-Net (ours) | $\underline{2.12 \pm 0.13}$ | $\underline{0.93 \pm 0.03}$ | $\mathbf{3.00 \pm 0.22}$ | $\mathbf{2.31 \pm 0.31}$ | $\underline{0.9465 \pm 0.0048}$ |

when comparing the error relative to the actual tumor volume, this assumption does not hold. Although the coefficients would imply a large tumor, the ground truth volume remains small due to anatomical barriers within the brain. These barriers act as non-differentiable boundary conditions, which may explain the observed errors.

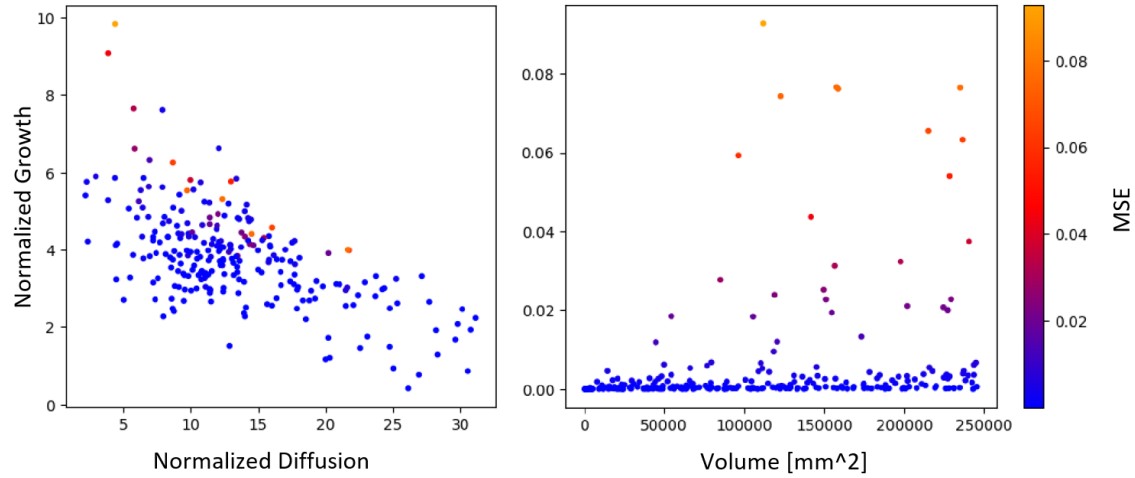

Figure 6: Error of GB Neural Surrogate. On the left, we show the ground truth normalized growth rate over the ground truth normalized diffusion rate. On the right, we show the MSE over the ground truth volume. Additionally, we color-code the MSE.

## Appendix D. Synthetic Data Generation

Synthetic data was generated using the TumorGrowthToolkit (Balcerak et al., 2023), which solves the Fisher-Kolmogorov equation on the atlas brain geometry. Tissue segmentations for white matter (WM) and gray matter (GM) were extracted from the atlas, with a diffusion ratio of $D_{GM} = 0.1 \cdot D_{WM}$. For each sample, we uniformly sampled the parameters within the following ranges:

- Diffusion coefficient: $D_{WM} \in [0.001, 10.0]$ mm$^2$/day

- Proliferation rate: $\rho \in [0.001, 10.0] \text{ day}^{-1}$

- Stopping volume: $V_{\text{stop}} \in [1000, 400000] \text{ mm}^3$

The stopping volume range was chosen to match tumor sizes observed in the BraTS dataset. Tumor seed locations were randomly placed within brain tissue (WM or GM). Each sample consists of the final tumor concentration field, the corresponding tissue segmentation, and the 5 normalized PDE coefficients $\theta = \{x, y, z, \mu_D, \mu_\rho\}$.

## Appendix E. Dice Comparison

In Figure 7 we showcase how the Dice score changes depending on the threshold chosen for the predicted continuous tumor concentration.

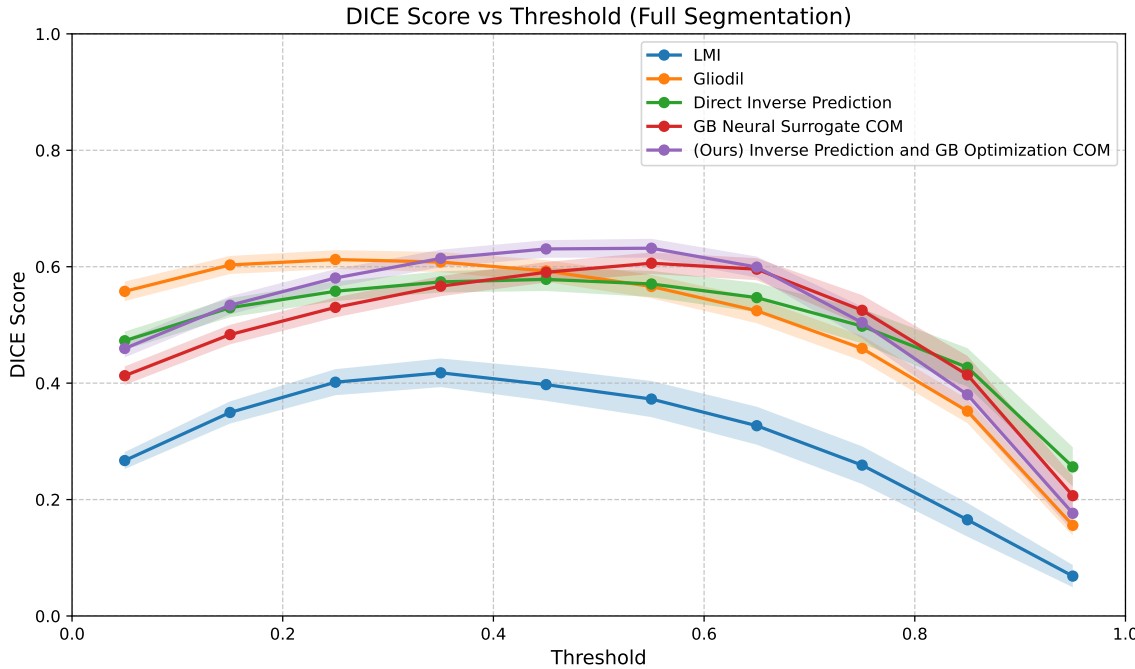

Figure 7: We show the Dice score over multiple thresholds for different methods on real patients. The tumor segmentation is compared to the thresholded predicted tumor concentration. Overall, we see that pure thresholding is not an ideal imaging function to describe the tumor segmentation. We find that our method performs comparably with GliODIL, while LMI (Ezhov et al., 2023) shows lower fit.

