# OpenReview forum: "Brain Tumor Growth Inversion via Differentiable Neural Surrogates"
_MIDL.io/2026/Conference — MIDL 2026 Poster_

### Official Review · Reviewer_XS4A · 2026-01-08

**Confidence:** 5
**Preliminary Rating:** 3
**Final Rating:** 4

**Summary:**

The paper presents a method for personalizing a biophysical brain tumor growth model (the classical 3D Fisher-Kolmogorov for GBM growth) using differentiable neural surrogates. These surrogates replace traditional numerical solvers for the identification of model parameters (using CMA-ES, BFGS in this work). The neural surrogate model proposed uses a ConvNeXt-enhanced U-Net architecture to learn the mapping between partial differential equation coefficients and simulated tumor concentrations.

The method is evaluated on 500 synthetic test cases, generated using the TumorGrowthToolkit, as well as on 75 patient cases from the BRATS dataset. Experimental results demonstrate that the approach effectively recovers model parameters while maintaining competitive accuracy. Additionally, a hybrid optimization strategy, combining direct inverse prediction with gradient-based refinement, is shown to enhance prediction quality. This improvement is assessed using metrics such as normalized cross-correlation and mean squared error.

**Strengths:**

The paper demonstrates several significant strengths:
- It introduces an innovative application of differentiable neural surrogates, enabling gradient-based optimization that substantially accelerates the identification of model parameters.
- A hybrid optimization strategy is proposed, integrating direct inverse prediction with gradient-based refinement to enhance prediction accuracy.
- The approach is validated through extensive experimentation.

**Weaknesses:**

In my opinion, here are the main weaknesses and limitations of the paper:
- The approach relies heavily on synthetic data, even for the 75 BRATS patients, where the best estimate is derived from a numerical solver. This reliance raises questions about the clinical relevance of the approach, as the practical utility of the inferred tumor-cell concentration for radiotherapy planning remains unclear. Specifically, the comparison between tumor segmentation and concentration in Figure 4 does not fully elucidate the added value for clinical decision-making.
- The sensitivity to initial conditions or local minima for gradient-based methods is not thoroughly explored. Although fixing the origin to the center of mass appears to improve results, it is unclear why starting from the solution of the direct inverse prediction is guaranteed to be close to the true optimum.
- The generalizability of the approach is limited by the training setup of the forward model. Factors such as the quality of the atlas and the sampling of the coefficients may constrain the model's ability to generalize effectively to diverse patient populations or different anatomical contexts.

**Detailed Comments:**

The paper is overall well written, and the authors mention that the code will be made available at a later date. However, to enhance readability, additional details could have been provided e.g. regarding the training of the models and the ranges of parameter variations. Furthermore, the regions or conditions where the approach fails are not adequately identified.

**Justification Of Final Rating:**

In light of the authors’ revisions, I have upgraded my evaluation of the article. The amendments enhance the clarity and thoroughness of the work’s description, while partially addressing the concerns raised in my initial review. The improvements contribute to a more robust presentation of the study’s methodology and findings, thereby warranting a higher assessment.

**Justification Of The Preliminary Rating:**

The approach presented in this paper is undeniably interesting, and the results clearly demonstrate a meaningful improvement over the compared methods. The advancements made in relation to the MIDL 2024 paper cited by the authors are substantial.
However, while these contributions are noteworthy, I find the paper to be borderline in terms of its overall impact. To warrant a more favorable assessment, I would have expected a deeper exploration of two critical aspects: first, a more thorough evaluation of the approach’s robustness and second, a clearer demonstration of its clinical relevance, such as how the inferred tumor-cell concentrations or parameters could concretely inform radiotherapy planning. Without such extensions, the paper, while technically sound, leaves some questions unanswered.

**Questions To Address In The Rebuttal:**

The work is interesting, but the approach would have benefited from a more comprehensive validation that enables a better assessment of its applicability beyond its current context and its relevance for clinicians.

---

> ### Author Response · Authors · 2026-01-25
>
> **Strengths:**
>
> We appreciate the recognition of using differentiable surrogates to enable efficient gradient-based inversion, the hybrid strategy combining direct inverse prediction with gradient-based refinement, and the extensive experimental validation.
>
> **Weaknesses:**
>
> 1. Limited clinical relevance due to heavy reliance on synthetic data.
>
> Thank you for raising this point. We agree that this work primarily focuses on synthetic data for solving the inverse problem. For clinical motivation, we extend the discussion and revise the explanation in Figure 4 to clarify the following two points.
>
> First, it has been shown that inversely derived growth and diffusion coefficients exhibit predictive power for overall survival and proliferation-related pathway activity, which is highly relevant for clinical decision-making (Metz et al., 2024).
>
> Second, several prior works have demonstrated improvements in radiotherapy planning using biophysical models (Balcerak et al., 2023). Consequently, our work focuses specifically on solving this inverse problem, reducing runtime orders of magnitude, which is critical for clinical applicability, while maintaining performance and reliability. To highlight this clinical application, we now also calculate and compare Dice score and find that our method performs comparable to the much slower state-of-the-art GliODIL method.
>
> 2. Sensitivity to initialization and local minima not sufficiently analyzed.
>
> We do not claim that the direct inverse prediction is guaranteed to be closer to the true optimum, but we found it to be a reasonable estimate (Table 1), which is the reason why we initialized the optimization with it. In prior studies, we found that the center of mass is a good predictor of tumor origin, and we also observed this in the described experiment. We assume this is due to the different test anatomy.
>
> To test whether our method is stable across different initializations of the growth and diffusion coefficients, we added a stability analysis to the Appendix. By investing in the loss landscape, we find indeed local minima where some of the optimization runs converge into, as mentioned by the reviewer. Thus, we show the benefit of initializing the optimization with the direct inverse prediction, validating our experimental findings.
>
> 3. Limited generalizability due to training and data assumptions.
>
> We agree that generalizability is constrained by the forward surrogate’s training setup, including atlas quality and coefficient sampling, which can limit transfer to diverse anatomies and populations. Therefore, we tested our approach on real patients with varying, unseen anatomy. We now explicitly added this as a limitation in the revised manuscript. However, to highlight the broad applicability of our method, we now also perform an additional comparison with different neural forward solver backbones (such as UNet, VNet) and find that our method works well with the different backbones – highlighting its flexibility and suitability also to other training setups.
>
> **Summary:**
>
> We thank the reviewer for the thorough and constructive feedback. In the revision, we expanded the robustness analysis. We also clarified the clinical motivation for infiltration-aware radiotherapy planning, explaining how concentration maps beyond the visible segmentation can inform target margins, and expanded the discussion and Figure 4 accordingly. We hope this makes robustness and clinical relevance more concrete.

---

> > ### Author Response · Authors · 2026-01-29
> >
> > Thank you again for your detailed and helpful review. We have revised the manuscript to further clarify the clinical motivation and to expand the robustness analysis, including additional stability experiments and a discussion of initialization effects. We would be thankful for any indication as to whether the revisions resolve your concerns or if further adjustments would be helpful.

---

### Official Review · Reviewer_qbRo · 2026-01-09

**Confidence:** 4
**Preliminary Rating:** 3
**Final Rating:** 4

**Summary:**

The paper presents a hybrid framework for estimating optimal PDE coefficients in biophysical brain tumor modeling. The approach consists of two stages: (1) a direct inverse prediction module, where a neural network produces an initial, physically plausible estimate of tumor parameters, and (2) a gradient-based refinement stage, in which this estimate initializes an optimizer that leverages a neural surrogate to further refine the parameters. The authors provide thorough and transparent baseline comparisons and demonstrate substantial improvements in computational efficiency, reducing optimization time from days to under one minute. In addition, the method is evaluated on real patient data, further supporting its practical applicability.

**Strengths:**

The primary strength of the paper is the proposed methodology’s ability to achieve substantially faster runtimes compared to conventional approaches while maintaining competitive performance metrics. Additionally, the analysis investigating the factors contributing to the slight performance gap relative to the conventional method is well executed and adds valuable insight into the model’s behavior.

**Weaknesses:**

While the paper has several strengths, it omits important methodological details that are necessary for reproducibility and for enabling future research. Specifically, the manuscript does not sufficiently describe the training procedure, the exact model architecture, or the loss functions used. In addition, details regarding the simulation dataset are lacking, making it difficult for readers to understand how the data were generated or to replicate the dataset. Providing these details would significantly improve the clarity and reproducibility of the work.

**Detailed Comments:**

Major Points:

(1) Please provide a detailed description of how the simulated dataset was generated, including the underlying assumptions, parameter ranges, and simulation pipeline.

(2) The manuscript would benefit from a comprehensive description of the model architecture, loss functions, and training procedure to ensure reproducibility.

(3) There is concern that the model may not be sufficiently regularized, potentially leading to overfitting of the inferred parameters and, consequently, inferior performance compared to CMA-ES. Please clarify the regularization strategies used and provide evidence to address this concern.

(4) Please justify the choice of tumor origin as the primary evaluation metric for the real patient analysis.

(5) It would be useful to explain why Dice score was not evaluated, particularly given that it is reported in related work such as Learn-Morph-Infer (Ezhov et al.).



Minor points:

(1) Figure 4 requires additional explanation, as it is unclear what comparisons the reader should focus on.

(2) Please clarify the rationale for using the older BraTS dataset (Menze et al., 2014) for simulation. Would the newer version (Baid et al., 2021) not provide a more realistic or robust basis for generating simulated data?

**Justification Of Final Rating:**

Thanks to authors for resolving all my concerns, I see a positive change in the manuscript regarding all details. I think overall its a good paper. While there is small degradation in performance, the overall runtime is significantly smaller.

**Justification Of The Preliminary Rating:**

Overall, this is a positive paper, but it has several gaps that need to be addressed. I have assigned a borderline score because there are aspects of the methodology and evaluation that could be improved or clarified. I am open to revising my score if these points are addressed with clear and well-justified responses.

**Questions To Address In The Rebuttal:**

I would like all major and minor points to be addressed.

---

> ### Author Response · Authors · 2026-01-25
>
> **Strengths:**
>
> We appreciate the reviewer’s encouraging comments. We are grateful for the recognition of our method's efficiency and competitive metrics compared to conventional approaches. We also thank you for acknowledging the depth and utility of our analysis regarding the factors contributing to the performance gap.
>
> **Weaknesses:**
>
> 1. Insufficient description of simulated data generation.
>
> Thank you for this point. We added a detailed description of the synthetic dataset generation in the revised manuscript, on all points mentioned.
>
> 2. Lacks detailed model and training description.
>
> We added a comprehensive description of the model architecture, loss functions, and the training procedure (including key hyperparameters and optimization settings) in the revised manuscript to ensure reproducibility. For transparency, we also published the code.
>
> 3. Concerns about insufficient regularization and overfitting.
>
> Our inversion process is inherently regularized by the problem structure. We do not optimize arbitrary patient-specific weights; rather, we optimize a low-dimensional coefficient vector $\theta$ strictly confined within physiologically plausible bounds. This constrained optimization acts as a strong prior, preventing the model from drifting into unrealistic regimes. Furthermore, we stabilize the process using robust initialization (direct inverse prediction) and a selection rule that retains the direct estimate if the optimization diverges. To demonstrate that this approach avoids overfitting, we report results on a held-out synthetic test set and, crucially, on 75 BraTS patients without any patient-specific fine-tuning. The fact that our method generalizes to real patient anatomy, which was not seen during training, strongly suggests that the parameters are robust and not overfitted.
> Regarding the learnable forward solver, we have extensively validated its stability on hold-out test data. We have added a new table in the Appendix that compares multiple forward architectures. We demonstrate that the surrogate model reliably replicates the underlying physics comparable to the numerical solver, while being orders of magnitude faster. Additionally, we added a stability analysis for the inverse optimization showcasing the impact of the initialization and the convergence behavior.
>
> 4. Justification of evaluation metric is missing.
>
> The tumor origin is not the primary evaluation metric on real patients; we mainly report image-space fit metrics (MSE, MAE, normalized errors, NCC) relative to the CMA-ES reference, and now also Dice score (see below). We discuss origin specifically because it is a key failure mode when transferring from atlas-based synthetic training to patient anatomies. Thus, we additionally include a COM-fixed variant to isolate its impact.
>
> 5. Missing justification for not using Dice score.
>
> Thank you for pointing this out. Originally, we did not report Dice because our inverse optimization is performed directly on continuous tumor-cell concentrations, and we compare against the concentration ground truth from the costly CMA-ES solver. Dice depends on a chosen imaging or thresholding function as done by Ezhov et al., which introduces an extra source of variability that is not central to the inverse problem.
>
> But we agree that for additional comparison, the widely used Dice is helpful. Thus, we added it to the comparison for real patients by using a simple threshold as an imaging function to compare to the tumor segmentation. Our method achieves a Dice score of 0.59, which is competitive with the significantly slower GliODIL method (0.61).
>
> **Minor:**
>
> 1. Figure 4, additional explanation.
>
> Thank you for pointing this out. We expanded the caption and in-text explanation for Figure 4 to clearly state the main message.
>
> 2. Unclear rationale for using older BraTS version.
>
> Thank you for pointing this out, and apologies for the mistake. We used the BraTS 2021 release for our experiments. We cited Menze et al. (2014) as the original BraTS benchmark paper, but we agree that the specific release citation should be explicit, and we corrected this in the revised manuscript to (Baid et al., 2021).
>
> **Summary:**
>
> Thank you for highlighting these concerns and for your detailed review. In the revised manuscript, we addressed all major and minor points by adding the missing implementation details, including the training procedure, model architecture specification, all loss terms, solver and optimization settings, and by expanding the description of the simulation dataset and its generation process to make the experimental setup reproducible. We also clarified the evaluation choices and figures. For exact implementation details, we have additionally released the code at https://github.com/jonasw247/brain-tumor-growth-inversion-via-differentiable-neural-surrogates. We hope the revised manuscript now presents the approach and evaluation in a sufficiently clear and well-supported manner.

---

> > ### Author Response · Authors · 2026-01-29
> > **Follow-up on revisions and additional experiments**
> >
> > Thank you again for your careful and insightful review. We have made substantial revisions to the manuscript, in particular by expanding the description of the simulation pipeline and providing full details on the model architecture, losses, and training procedure. We would very much appreciate your feedback on whether these changes adequately address your concerns, or if further clarification would be useful.

---

### Official Review · Reviewer_jxsr · 2026-01-10

**Confidence:** 4
**Preliminary Rating:** 3
**Final Rating:** 4

**Summary:**

This study proposes a methodology to solve inverse problems in personalized medicine simulation through the implementation of a neural surrogate which transforms a non-differentiable biophysics model of tumoral growth into a differentiable system with gradient descent optimization for the efficient inverse calibration of patient specific parameters. The proposed method also demonstrated superior performance on a synthetic and a publicly available patient dataset.

**Strengths:**

1. The work proposes a differentiable neural surrogate framework for the inverse calibration of PDE-based tumor growth models, providing an important contribution to the field of personalized medical modeling.
2. The authors conducted a detailed analysis of the failure modes, limitations of the parameter space, and behavior outside of the distribution, and applied this to an evaluation of real patient data.
3. The method has been shown to improve synthetic data produced by the TumorGrowthToolkit, and to also perform well on real patient data from the BraTS dataset, demonstrating its practical application.

**Weaknesses:**

1. The proposed method uses ConvNeXt–UNet structure but there are no extensive comparisons with classic UNet, VNet, UNet++ or other comparative baselines. The authors should provide a more detailed comparison for the proposed method.
2. It is not entirely clear how the data set BraTS was utilized. The authors mention that 75 patients from the BraTS data set were used in their experiments but they do not indicate which release of the BraTS data set the patients were obtained from either. Furthermore, it is unclear why the authors have limited the evaluation to the tumor subtypes (Glioma, Meningioma, or Metastasis) present in the BraTS data set and not included the full complement of tumor subtypes available in BraTS.
3. The authors have provided very little detail regarding their numerical solver and how they addressed the implicit and explicit terms in the PDEs. At this time, it is not possible to assess the validity and stability of the forward model based on the information contained in this manuscript.
4. The authors have not provided sufficient clarity regarding the description of their gradient-based neural surrogate. It is not clear how the gradient is calculated from the surrogate and the steps leading to L-BFGS optimization. It would be great if the authors can provide the pseudo-code for their optimization step.

**Detailed Comments:**

Brain volume is divided into three categories: white matter, gray matter, and cerebrospinal fluid. Please specify how you divided the brain volume into WM, GM, and CSF (SynthSeg or another segmentation tool).

**Justification Of Final Rating:**

The authors have adequately addressed the key concerns raised during the review process through their rebuttal and revisions. They have added additional baselines and clarified several aspects of the paper. Accordingly, I update my final recommendation to `weak accept'.

**Justification Of The Preliminary Rating:**

While the proposed approach has numerous strengths including the implementation of a differentiable surrogate framework for inverse tumor growth modeling, a lack of methodological clarity, lack of description of solvers used, and lack of comparison against existing methods reduce confidence in how robust and broadly applicable the proposed approach will ultimately prove to be.

**Questions To Address In The Rebuttal:**

Please check the weakness section.

---

> ### Author Response · Authors · 2026-01-25
>
> **Strengths:**
>
> We appreciate the recognition of our differentiable neural surrogate framework and its relevance for personalized tumor modeling. We also thank the reviewers for highlighting the detailed analysis of failure modes and out-of-distribution behavior, which we included to transparently assess robustness. Finally, we are grateful for the acknowledgment of the strong performance on both synthetic TumorGrowthToolkit data and real BraTS patient data, demonstrating the practical applicability of our approach.
>
> **Weaknesses:**
>
> 1.	Missing baseline comparisons:
>
> We thank the reviewer for this suggestion. We have added a comprehensive ablation study in the Appendix comparing our ConvNeXt-U-Net against a standard U-Net, U-Net++, and V-Net. The results show that while V-Net performs slightly better on absolute MSE, the ConvNeXt-UNet offers an overall superior, balanced performance across metrics. This confirms that the forward surrogate accuracy is robust across varying backbones, validating that the primary contribution, the inverse optimization framework, is not dependent on a specific architecture, but rather a broadly applicable methodological contribution.
>
> 2.	Unclear BraTS usage and selection criteria:
>
> We clarify in the revised manuscript that BraTS release 2021 was used, including the relevant citation.  We also clarified that the 75 patients were selected randomly. The evaluation is limited to the tumor entities supported by the underlying biophysical tumor growth model, which is designed for infiltrative IDHwt glioblastoma growth. We explicitly state this in the revision, but acknowledge that studies into the specific growth patterns of other tumor entities are an attractive future research direction.
>
> 3.	Insufficient detail on the numerical solver and PDE implementation:
>
> We thank the reviewer for raising this point. The numerical solver is based on the open source TumorGrowthToolkit implementation, which has been validated and used in several prior studies. It uses an explicit finite-difference scheme with a forward Euler update and a conservative time step for stability. We describe the details in the revised manuscript in a separate section in more detail.
>
> 4.	Unclear gradient computation and optimization procedure:
>
> We agree that some details are missing for the optimization. We have expanded the description of how gradients are computed via the neural surrogate and used within the L-BFGS optimization, including pseudo-code to explicitly illustrate the optimization procedure in the revised manuscript.
>
> 5.	Unclear brain tissue segmentation method:
>
> Thank you for pointing out this important aspect. We have clarified in the revised manuscript that we generate the WM, GM, and CSF masks by deformably registering a brain atlas to the patient image. The reason for this is that we want to recover the anatomy close to the healthy (tumor-free), but already deformed (due to mass effect) brain. We found that standard segmentation tools such as SynthSeg and Atropos do not provide sufficient tissue segmentation in the tumor region.
>
> **Summary:**
>
> We thank the reviewer for the thorough and constructive feedback. We addressed the raised weaknesses by clarifying the methodology, adding solver and optimization details (including pseudo-code), and strengthening the relevant comparisons. We hope these clarifications address the key concerns and will lead to an updated assessment.

---

> > ### Author Response · Authors · 2026-01-29
> > **Follow-up on revisions and additional experiments**
> >
> > Thank you again for your detailed and constructive feedback. We have carefully revised the manuscript to address all raised points, including added architectural comparisons and detailed solver and optimization descriptions. We would greatly appreciate your feedback on whether these revisions sufficiently address your concerns or if further clarification would be helpful.

---

### Author Rebuttal · Authors · 2026-01-25

**Rebuttal:**

We thank all reviewers for their careful evaluation and constructive feedback. We are encouraged that they find our work to be an "important contribution to the field of personalized medical modeling" (Reviewer jxsr), recognize our "proposed methodology’s ability to achieve substantially faster runtimes compared to conventional approaches while maintaining competitive performance metrics" (Reviewer qbR0), and that our "approach is validated through extensive experimentation" (Reviewer XS4A).

Across reviews, the main concerns related to methodological clarity, reproducibility, solver and optimization details, architectural comparisons, robustness, and clinical relevance. We have addressed these points in the revised manuscript.

First, we expanded the methodological description substantially. This includes a detailed explanation of the numerical solver, discretization, and stability conditions, as well as a clear description of gradient computation through the frozen surrogate. We added pseudo-code for the L-BFGS optimization and provided relevant details on model architecture, loss functions, hyperparameters, and synthetic data generation and released our code.

Second, regarding architectural choices, we clarified that the forward surrogate backbone is a design choice rather than a baseline for the inverse problem, which is the core contribution. Nevertheless, we added an architectural ablation comparing ConvNeXt–UNet with classic alternatives and referenced prior work that evaluates transformer-based surrogates.

Third, we clarified the use of the BraTS dataset by explicitly stating the release used, the random patient selection, and the restriction to infiltrative tumor entities supported by the underlying biophysical model.

Fourth, we strengthened the robustness analysis by expanding the failure evaluation and clarifying the role of initialization. The proposed hybrid strategy, which combines direct inverse prediction with gradient-based refinement, proves beneficial.

Finally, we clarified the clinical motivation. Our goal is infiltration-aware radiotherapy planning, and inversely inferred growth and diffusion parameters have already demonstrated clinical relevance themselves. We therefore added the Dice metric and expanded the discussion and visual explanations accordingly.

We believe these revisions substantially improve the clarity, robustness, and transparency of our work, and address the reviewers’ key concerns.

**Supporting Material:**

/attachment/75edcadb545e144235f9e65c67351fcbb323daf8.pdf

---

### Comment · Area_Chair_jNjZ · 2026-01-30
**Update final rating**

Dear reviewers,

This paper is borderline. Please take the authors’ rebuttal into consideration and update your final rating by February 1. Thanks for your time and effort!

Best,
MIDL AC

---

### Meta-Review · Area_Chair_jNjZ · 2026-02-06

**Recommendation:** Accept (Poster)
**Confidence:** 5

**Metareview:**

All reviewers’ recommendations leaned toward acceptance after the authors’ rebuttal. The proposed idea of using differentiable neural surrogates to approximate PDE-based tumor growth models for speeding up gradient-based optimization is interesting and has the potential to be valuable for medical applications.

---

### Decision · Program_Chairs · 2026-02-13

Accept (Poster)